# Common and Rare Hematological Manifestations and Adverse Drug Events during Treatment of Active TB: A State of Art

**DOI:** 10.3390/microorganisms9071477

**Published:** 2021-07-09

**Authors:** Maria Letizia Minardi, Ilenia Fato, Francesco Di Gennaro, Silvia Mosti, Annelisa Mastrobattista, Carlotta Cerva, Raffaella Libertone, Annalisa Saracino, Delia Goletti, Enrico Girardi, Massimo Andreoni, Fabrizio Palmieri, Gina Gualano

**Affiliations:** 1National Institute for Infectious Diseases “L. Spallanzani” IRCCS, 00149 Rome, Italy; marialetizia.minardi@inmi.it (M.L.M.); Ilenia.fato@inmi.it (I.F.); silvia.mosti@inmi.it (S.M.); a.mastrobattista@inmi.it (A.M.); Carlotta.cerva@inmi.it (C.C.); Raffaella.libertone@inmi.it (R.L.); delia.goletti@inmi.it (D.G.); enrico.girardi@inmi.it (E.G.); Fabrizio.palmieri@inmi.it (F.P.); gina.gualano@inmi.it (G.G.); 2Clinic of Infectious Diseases, Policlinico Tor Vergata, V. Montpelier 1, 00133 Rome, Italy; andreoni@uniroma2.it; 3Department of Biomedical Sciences and Human Oncology, Clinic of Infectious Diseases, University of Bari “Aldo Moro”, 70124 Bari, Italy; annalisa.saracino@uniba.it

**Keywords:** tuberculosis, adverse events, hematopoietic system, anemia, rare hematologic disorders

## Abstract

Background: Tuberculosis (TB) can seriously affect the hematopoietic system, with involvement of both myeloid and lymphoid cell lines as well as plasma components. These hematological changes act as a marker for the diagnosis, prognosis and response to therapy. Methods: We searched PubMed, Scopus, Google Scholar, EMBASE, Cochrane Library and WHO websites from 1950 to May 2021 for papers on the interaction between TB and common and rare hematological manifestation. Results: Hematological reactions in patients with TB are possible in both young and old women and men but seem more frequent in the elderly, and they can be predictors of both diagnosis and worse outcome for TB, regardless of whether it is pulmonary, extra pulmonary or miliary. Even anti-TB therapies can cause hematological adverse events, among which some are serious and rare and can compromise the patient’s recovery pathway to completing treatment. Conclusion: Hematological screening and follow-up, including complete blood count and coagulation, are always necessary both at the diagnosis of TB and during antitubercular treatment in order to monitor hematological parameters. Short therapy regimens for multidrug-resistant TB (MDR-TB) may also be useful for reducing hematological toxicity, especially in contexts where this cannot be monitored. Close monitoring of drug interactions and hematological adverse events is always recommended.

## 1. Introduction

Despite all the efforts made, tuberculosis (TB) remains one of the top 10 causes of death worldwide and the leading cause of death from a single infectious agent. In fact, globally, it has been estimated that 10.0 million people had TB in 2019, with around 1.43 million deaths [1]. *Mycobacterium tuberculosis*, mainly affecting the lungs but also other organs, is the cause of TB [2]. The great variability in the clinical presentation of TB poses a diagnostic and therapeutic challenge for physicians. In addition, TB infection can seriously affect the hematopoietic system during its course, with involvement of both myeloid and lymphoid cell lines and plasma components [3]. Various hematological abnormalities in association with TB have been reported in the literature, and these hematological changes may act as a marker for the diagnosis, prognosis and response to therapy [3]. However, there is limited knowledge about the prevalence of the effects of hematological abnormalities and adverse events of antituberculous treatment on various hematological parameters.

The aims of this study were to review the literature concerning hematological manifestations in TB patients in order to identify a relationship via which their diagnostic and prognostic significance could be detected and to monitor the eventual adverse events during antituberculous therapy.

## 2. Materials and Methods

We conducted a search on PubMed, Scopus, Google Scholar, EMBASE, Cochrane Library and WHO websites (http://www.who.int, accessed on 20 May 2021) starting from 1950 to May 2021 in order to identify articles discussing the interaction between TB and common or rare hematological manifestation. We included all articles addressing epidemiology, physiopathology, risk factors, clinical features, screening and diagnosis, treatment and management.

## 3. Blood Count Impairment in Active TB

### 3.1. Anemia

Anemia is one of the most common laboratory test abnormalities seen in TB [4]. The prevalence of anemia in patients with diagnosis of TB ranged from 32% to 94% in studies conducted in various countries [4] and can be associated with increased mortality and poor response to treatment [5].

Anemia has been considered a risk factor for the onset of infectious diseases and especially TB [6], as it increases individual susceptibility to such onset by reducing the immune response. Anemia is generally present at the time of diagnosis of TB. The most frequent form is a normochromic normocytic anemia [4] followed by microcytic anemia. In a previous retrospective study, authors observed that anemia is associated with more severe forms of TB, suggesting that it could be a biomarker of the severity of the disease [4], with the meningeal and disseminated forms accounting for the more severe forms of anemia as showed in Table 1.

Many different types of blood conditions, such as anemia, folate deficiency and sideroblastic disease, can affect TB patients [3]. A strong, positive association of non-iron-deficient anemia with TB recurrence, mortality, and HIV disease progression suggests that factors other than iron deficiency also contribute to the association of anemia with poor clinical outcomes. TB induces a systemic inflammatory response that stimulates the synthesis of hepcidin from hepatocytes and macrophages [7], which is the main factor in the iron metabolism. Hepcidin regulates absorption of iron: in inflammation, its concentration increases; therefore, the serum level of iron decrease, and this process could be responsible for anemia. Macrocytic anemias due to deficiency of folates or vitamin B12 have been described in the literature [8], mainly associated with malnutrition, an increase in folate consumption during the inflammatory phase of the disease, malabsorption syndrome due to intestinal and especially ileal localization of the disease [8]. Vitamin B12 deficiency is rarely reported despite the ileocecal region being the most common site for abdominal TB. Another factor to be considered is that anemia can also occur in the early phase of the treatment of TB after normal hematopoiesis reestablishment. As inflammation decreases during treatment, iron therefore becomes available for hematopoiesis, causing an increase in reticulocytes [5,6,7].

In a study conducted in Dar-es-Salaam, anemia at the beginning of TB treatment was strongly associated with delayed sputum smear conversion in sputum-positive TB patients, with a dose–response effect. Anemic patients were three times more likely to have a sputum-positive smear as than were non-anemic patients at two months, with the risk of sputum-positive smear results increasing with severity of anemia [9]. This is a factor to which the clinician must pay attention, as persistent sputum smear positivity during treatment is a predictor of adverse treatment outcome [9].

### 3.2. White Blood Cell Disorders (WBC)

Hematological abnormalities involving the white blood cells can be associated with TB and include leukopenia, neutropenia, lymphocytopenia, monocytopenia, leukocytosis, neutrophilia, lymphocytosis and monocytosis, and pancytopenia (mainly in miliary TB) [3]. In a Chinese cross-sectional and follow-up study considering leukopenia in patients with TB, female gender, older age, duration of previous anti-TB treatment for more than 6 months [6] and use of antibiotics were risk factors for leukopenia. The prevalence of leukopenia in TB patients newly treated was 10.4%, while that of the previously treated TB patients was 9.1% [10].

Neutrophils can be also affected, with neutrophilia being more common than neutropenia [10], and both parameters return to normal following successful treatment. In rare and more severe cases, a leukemoid reaction can occur, mimicking the appearance of acute leukemia. Neutropenia is rarer and is due to the suppression of granulopoiesis by activated T cells; folate and vitamin B12 deficiency; marrow fibrosis or dysfunction due to localization of disease; or splenic sequestration [5].

### 3.3. Platelet Disorders

TB is associated with thrombocytosis, which is related to the intensity of the inflammatory response; the extent of thrombocytosis can be monitored as an inflammatory marker, such as the erythrocyte sedimentation rate and serum C-reactive protein [11,12]. It is thought that thrombocytosis is related to increased levels of interleukin 6, which is responsible for the hypercoagulability state, deep vein thrombosis and thromboembolism [13].

Thrombocytopenia is more common in patients with miliary TB, while thrombocytosis is more common in patients with pulmonary TB [14]. Most patients with disseminated/miliary [13] TB show severe anemia, peripheral monocytopenia and bone marrow histiomonocytosis, and this is due to the formation of bone marrow granulomas. Hypersplenism, histiocytic hyperplasia, maturational arrest, and infiltration of the bone marrow by caseating or non-caseating granulomas are considered causes of pancytopenia in disseminated or extrapulmonary TB [12]. The hemogram returns to normal after specific therapy [12,13]. Some authors suggest investigating thrombocytosis with negative acid-fast staining (AFS) as a potential indicator of TB infection. Immune thrombocytopenia (ITP) is an autoimmune condition that results in isolated thrombocytopenia associated with possibly lethal hemorrhage; it can also be triggered by infectious and non-infectious conditions. Secondary ITP associated with TB has rarely been described in the literature [13].

## 4. Rare Hematological Manifestations in Active TB

### 4.1. Henoch–Schönlein Purpura

Henoch–Schönlein purpura (HSP) is a vasculitic syndrome characterized by the eruption of diffuse urticarial plaques and palpable purpura mainly on the lower extremities, abdominal pain, joint pain, and renal involvement, evidenced by proteinuria and/or hematuria [15]. This is an extremely rare condition, seen in several virus and bacterial infections and in patients with pulmonary TB. There are only a few reported cases, suggesting that it is difficult to make a definitive diagnosis when HSP presents as an initial manifestation of pulmonary TB [15]. Besides a mandatory criterion, such as palpable purpura (not thrombocytopenia), the latest diagnostic criteria of HSP include at least one of the following manifestations: diffused abdominal pain, leukocytoclastic vasculitis with predominant IgA deposits on skin biopsy, acute arthritis, arthralgia in any joint and renal involvement as evidenced by proteinuria and/or hematuria [16].

Pulmonary TB presenting with HSP as an initial manifestation is not common and consequently difficult to clinically diagnose and manage [16]. When an adult patient presents with HSP, it is important to consider the possibility of TB to avoid misdiagnosis and delayed treatment.

### 4.2. Pancytopenia

Pancytopenia is a hematological abnormality due to bone marrow localization of TB in extra-pulmonary TB [14]. Diagnosis is challenging, but rapid diagnosis and commencement of treatment are essential, as observed by Alghamdi et al. [14]. Several factors are considered to cause pancytopenia in disseminated or extrapulmonary TB (EPTB), including hypersplenism, histiocytic hyperplasia, maturation arrest and infiltration of the bone marrow by granulomas with subsequent fibrosis [17]. The clinical presentation includes fever, weight loss, pallor and massive splenomegaly with pancytopenia; in most cases, the diagnosis can be made only after bone marrow biopsy [1]. After starting treatment, the bone involvement decreases and the blood count improves; thus, the timing of TB treatment is very important.

### 4.3. Immune Hemolytic Anemia

To date, few cases of hemolytic anemia in patients with disseminated TB have been described [18]. Autoimmune hemolytic anemia (AIHA) is a rare condition associated with many infectious diseases and TB. Some studies have shown that injection of bacilli or their derivates is able to induce hemolytic anemia, pancytopenia and myelofibrosis in small animals [18,19]. In humans, the bacilli may cause marked proliferation of the reticuloendothelial tissues, with immune hematologic disorders. 

Glucocorticoids are usually the first line of treatment in AIHA with gradual tapering after the first days of treatment and dose adjustment in relapses. In resistant cases, intravenous immunoglobulins or immunosuppressive drug administration can be necessary [18,19]. In cases of hemolytic anemia associated with infection, immediate initiation of antimicrobial therapy is very important and may be lifesaving.

### 4.4. Myelofibrosis

Myelofibrosis, also known as myelosclerosis, is a rare hematologic disorder characterized by extensive fibrosis of the bone marrow [20]. Whether TB stimulates a secondary fibrotic reaction or develops in patients with preexisting myeloproliferative disorders is still debatable. Pancytopenia, secondary to the myelosuppressive effects of TB, has been supported by cases where recovery of peripheral blood counts with antituberculous therapy has been accepted as evidence of a normal bone marrow [20]. TB frequently raises the possibility of myelofibrosis among non-hematopoietic diseases, so clinicians must carefully monitor hospitalized patients with preexisting chronic myeloproliferative disorders. Some authors have reported greater severity of TB and exacerbation of myelofibrosis in patients treated with ruxolitinib, a kinase inhibitor that is widely used to treat myelofibrosis and polycythemia [21].

### 4.5. Hemophagocytic Lymphohistiocytosis

Hemophagocytic lymphohistiocytosis (HLH) is an uncommon, potentially fatal hyperinflammatory syndrome that may rarely complicate the clinical course of disseminated *Mycobacterium tuberculosis* (MTB) [22]. HLH should be considered in the differential diagnosis in patients with TB who present with cytopenias, organomegaly and coagulopathy. It is associated with cytokine storm and inflammation, which seriously affect quality of life. HLH is rare, with a lower incidence in adults than in children, and it usually presents secondary to cancer, infections, autoimmune and other diseases [22]. Among infectious diseases, secondary or acquired, HLH is commonly associated with viral infections, such as Epstein–Barr virus and cytomegalovirus, and bacterial infections. Secondary HLH can also be associated with TB-HLH. Hemophagocytic lymphohistiocytosis is a severe complication of TB infection and has a high mortality [23]. For patients with fever of unknown origin (FUO), HLH-related clinical manifestations sometimes present before the final diagnosis of TB.

### 4.6. Leukaemoid Reaction

Leukemoid reaction is a reversible hematological reaction defined as a response to severe infection, burns and other conditions associated with a high leukocyte count and smear blood films that resemble those seen in leukemic or sub-leukemic manifestations [24]. It is defined as a leukocytosis exceeding 50,000 cells/mL accompanied by an increase in neutrophil precursors, and the count of leukocytes rarely exceeds 60 × 109/L.

Due to alterations in the blood, similar to cases of hematological neoplastic disorders, it is necessary to distinguish it from leukemia [24]. Leukemoid reaction is usually apparent with many clinical signs. Leukocytosis associated with leukemoid reaction is transient and may return to normal when the causes are removed. Leukemoid reaction classification depends on the course of the disease and the development of hematopoietic irritation into myeloid (neutrophilic), leukemoid reaction, lymphoid leukemoid reaction, eosinophilic leukemoid reaction, monocytic leukemoid reaction and basophilic leukemoid reaction [25]. Leukemoid reactions resembles chronic myeloid leukemia or chronic neutrophilic leukemia; they may take place in TB, septicemia and acute megaloblastic anemia during pregnancy. These hematologic manifestations rarely occur, usually in patients with extrapulmonary manifestations or disseminated TB and more frequently in older males [24,25].

### 4.7. Hemophagocytic Syndrome

Hemophagocytic syndrome (HS) is a rare disorder of the immune system caused by the loss of the immunoregulatory capacity, characterized by fever, lymphadenopathy, hepatosplenomegaly, cytopenia and hyperferritinemia [26]. Infectious triggers described differ in each country, suggesting a specific genetic background and epidemiology. HS may be underdiagnosed due to variable clinical presentations, delaying prompt treatment and contributing to the high mortality rates [26]. TB is recognizable among rare causes of this syndrome, especially in EPTB. Diagnostic criteria of HS are as follows: fever, splenomegaly, cytopenia (at least 2 lines), hypertriglyceridemia and/or hypofibrinogenemia and hemophagocytosis in bone marrow, the spleen or lymph nodes. In a low-resource setting, even limited activation of the diagnostic pathway for disseminated TB can expedite empiric treatment of disseminated TB, thereby saving lives [26,27]. Conversely, a delay in identifying HS as the underlying cause of either disseminated TB or TB-related pancytopenia might be fatal [27].

### 4.8. Disseminated Intravascular Coagulation

The development of disseminated intravascular coagulation (DIC) is an extremely rare life-threatening manifestation of TB. Its pathogenesis is still not clear, probably due to mycobacterial endo- or exo-toxins, capable of initiating the clotting cascade, or protein release into circulation during bacteremia [28]. The incidence of DIC is high in patients with miliary TB, as reported by Stein et al., who reviewed 13 cases of miliary TB associated with high mortality [29], while other authors described the case of two TB patients, the first with splenic and mesenteric lymph node TB and the second with miliary TB involving the lungs and liver [30]. Both showed a low blood fibrinogen concentration, thrombocytopenia and other blood disorders, such as hemorrhagic syndrome.

### 4.9. Thromboembolism

Thromboembolism is a complication with a higher prevalence in patients with TB than in the general population, as confirmed in many meta-analyses (1.2% to 2.7%), and this could be due to the presence of venous stasis, endothelial lesions and hypercoagulability [31]. TB can be responsible for hypercoagulability as a result of chronic inflammation, which can cause an increase in the plasma level of factor VIII, fibrinogen and the plasminogen activator inhibitor as well as a reduction in protein C and antithrombin III. Various studies demonstrate the higher prevalence of thromboembolism in TB patients [32,33] than in the general population; Ha et al., for example, studied 7905 TB patients at Seoul National University Boramae Medical Center from January 2000 to March 2015, reporting an incidence of thromboembolism that was higher than that in the general population (approximately 0.1%) [32]. Higher incidence (about 2%) was reported in other papers [33,34].

No association between clinical manifestation and TB has been demonstrated. In a recent Canadian study on 240 TB patients [35] and in large retrospective data from China, the incidence of thromboembolism in pulmonary TB was 4.2% and 2.8%, respectively [36]. The prevalence was higher (31%) in a cohort of patients with autoimmune diseases, but the higher prevalence in this study could be explained by the conjunction of two proinflammatory conditions in patients with limited mobility [37]. Another retrospective review was performed by Sharif-Kashani et al. in 3293 TB patients. The authors showed that venous thromboembolic events are not rare in hospitalized TB patients, with an incidence of 0.94% [38]. Furthermore, a single-case report showed that inferior vena cava thrombosis and pulmonary thromboembolism can occur independently of age, representing a hematological condition that should always be considered in patients with TB [39,40].

## 5. Hematologic Adverse Event Associated with TB Treatment

### 5.1. Isoniazid

Isoniazid is a synthetic derivative of nicotinic acid with anti-mycobacterial proper-ties, commonly used in antitubercular therapy. Hematologic side effects of isoniazid have rarely been described. Pure red cell aplasia was described for the first time by Goodman in 1964, and other recent publications confirm that it is a rare condition [41]. Furthermore, due to the inhibition of δ-aminolevulinate synthase-214, isoniazid can also be responsible for sideroblastic anemia. The frequency is rare, and only eight cases are described in the literature, probably due to the generally medication co-prescribed with pyridoxine [42]. Very rare side effects as described in some case reports are isoniazid-induced agranulocytosis and thrombocytopenia [41,42,43].

### 5.2. Rifamycins (Rifampicin, Rifabutin and Rifapentine)

The rifamycins are bactericidal antibiotics inhibiting bacterial DNA-dependent RNA polymerase and suppressing RNA synthesis; they are usually used as first-line combination therapy for TB. Rifampicin, which is the most commonly used rifamycin, is associated with various hematological side effects. Rifampicin-induced thrombocytopenia was reported both in pulmonary and extrapulmonary TB [44], while TTP was described in a case report of a patient under treatment for pulmonary TB and in a female treated for latent tuberculosis [45]. In addition, Van Assendelft and colleagues studied the frequency of leukopenia in two groups of patients treated with two different preparations of rifampicin [46].

An uncommon known side effect associated with drug administration is DIC [47]. Intermittent administration of rifampicin has shown to determine a rare immunoallergic reaction, which is IgG and IgM mediated, against erythrocytes; platelets (PLTs); and other target cells expressing blood antigen I, including renal tubular epithelial cells [47]. Sousa et al. identified only 13 previously reported cases of rifampicin-induced DIC [48], underlining how this side effect is very rare. Authors described a rare case of agranulocytosis as a side effect in a chronic renal failure patient [49]. Furthermore, two cases of hemorrhage associated with vitamin K deficiency in pregnant women and newborns related with rifampicin therapy were reported [50]. The mechanism responsible for rifampicin-induced hemolysis is the existence of rifampicin-dependent antibodies directed at I antigens, located at the surface of red blood cells. First reports on rifampicin-related hemolysis were in patients receiving intermittent high-dose treatment, suggesting that these conditions can be a prerequisite for the increased formation of rifampicin-dependent autoantibodies. Only few cases of intravascular hemolysis related to daily drug administration have been reported to date [51]. Table 2 summarize common and rare hematologic adverse event due to TB treatment.

As rifapentine was approved for medical use in 1998, various hematological side effects associated with its administration have been observed, such as acute porphyria, aplastic anemia, autoimmune hemolysis, agranulocytosis, leukemoid reaction, leukopenia, thrombocytopenia and TTP, but little evidence is reported in the literature.

Rifabutin is similar in structure and activity to rifampicin and rifapentine; it is largely used in the prevention of the *M. avium* complex (MAC) in advanced HIV infection and in those HIV patients who do not tolerate rifampicin.

The main hematological adverse event reported for rifabutin is leukopenia, as reported by Chitre et al. It is currently unknown whether the leukopenia is dosage related or idiosyncratic [52]. Apseloff et al. described severe neutropenia among several healthy volunteers receiving therapeutic doses of rifabutin [53] as the most frequently reported adverse event after administration of rifabutin, occurring in 33 of 50 subjects and ranging from mild to potentially life threatening; it has also been demonstrated that rifabutin caused a significantly greater decrease in absolute neutrophil counts than did rifampicin [54]. 

### 5.3. Ethambutol

Ethambutol is an antibiotic with bacteriostatic, antimicrobial and antitubercular properties, and it interferes with the biosynthesis of arabinogalactan, a major polysaccharide of the mycobacterial cell wall. It has been associated with various hematologic side effects, although rare, affecting mainly elderly people and mostly females. Some authors described ethambutol-induced hemolytic anemia in a 71-year-old patient with a diagnosis of reactivated silico-TB [55], while other authors reported agranulocytosis and thrombocytopenia during treatment [56,57]. In addition, some authors reported neutropenia and eosinophilia in an elderly patient with reactivation of pulmonary TB as a side effect [58].

### 5.4. Pyrazinamide

Pyrazinamide is a synthetic pyrazinoic acid amide derivative with bactericidal properties, and it is particularly active against slowly multiplying intracellular bacilli. In the literature, there are a few cases of hematologic adverse events due to pyrazinamide. Kant et al. described thrombocytopenia both in a 20-year-old patient with tubercular meningitides and in a 75-year-old patient with pulmonary TB [59] and in association with sideroblastic anemia [60]. Megaloblastic anemia is a very rare event that may occur in elderly patients [61].

### 5.5. Streptomycin

Streptomycin (SM) is an aminocyclitol glycoside antibiotic that inhibits the initiation of protein synthesis discovered in 1943, and it was the first drug to be used in the treatment of TB in 1948 [62].

SM-associated eosinophilia has been described [63]. There are studies regarding hemolytic anemia due to streptomycin use, such as that of Nachman et al., who reported a 49-year-old female with pulmonary TB [64]. Other hematologic abnormalities found include leukopenia and thrombocytopenic purpura [62,63,64].

### 5.6. Amikacin

Amikacin is an aminoglycoside antibiotic used in the treatment of TB. The only reported hematologic side effect of amikacin is drug rash with eosinophilia and systemic symptoms (DRESS syndrome), a severe allergic syndrome characterized by generalized skin rash, blood eosinophilia and multiple-organ failure induced by drug-specific T cells. This case was reported by Bensad et al. in a 42-year-old male who was treated for septic arthritis [65].

### 5.7. Cycloserine

Cycloserine is a broad-spectrum antibiotic used as a second-line agent for the treatment of drug-resistant tuberculosis. It has shown many, although infrequent, hematological side effects. Vitamin B12 deficiency, folic acid deficiency, megaloblastic anemia, sideroblastic anemia and folate deficiency were described by Klipstein et al. in two patients and in a groups of 40 control subjects and 120 patients with pulmonary TB. Decreased serum folate concentrations were more frequent in patients receiving both cycloserine and isoniazid than in those being treated with other drugs [66].

### 5.8. Para-Aminosalicylic Acid

Para-aminosalicylic acid (PAS) is a bacteriostatic chemotherapeutic agent used in the therapy of tuberculosis caused by strains of the mycobacteria resistant to other anti-TB drugs or if the patient is intolerant to other drugs. After administration of PAS, many hematological side effects were observed. Thrombocytopenia has been reported both in adult patients with pulmonary TB [67] and in children [68]. Rab and colleagues described a case of severe and rare agranulocytosis due to PAS in a 19-year-old patient with pulmonary TB [69]. Hypersensitivity reactions to PAS occurred in 0.3 to 8% of persons receiving the drug, generally in about a month but occasionally as late as five months after the beginning of therapy. Eosinophilia is quite common and may reach a level of 40% of patients [70]. Acute hemolysis has been reported in two cases attributed to the use of PAS solution containing meta-aminophenol and other phenolic substances [71]. Rare cases of hemolytic anemia [72] and methemoglobinemia were described in both young and elderly pulmonary TB patients [70,72].

### 5.9. Levofloxacin and Moxifloxacin

Fluoroquinolones (FQ) are among the most important group of drugs for the treatment of MDR-TB. Few cases of hematologic reaction due to levofloxacin were reported [73]. Thrombocytopenia, pancytopenia, microangiopathy and hemolytic anemia are adverse events related to levofloxacin administration [73,74,75].

Two MDR TB patients manifested TTP after moxifloxacin administration with a mechanism poorly understood and probably secondary to hypersensitivity reaction [76]. Drug-induced thrombocytopenia is thought to occur via two mechanisms: impaired platelet production or increased platelet destruction (the latter typically occurs more often). While not completely explained, immune-mediated platelet destruction occurs when the drug interacts with platelets or antibodies to expedite the clearance of platelets [77].

### 5.10. Clofazimine

Clofazimine (CFZ) is a fat-soluble riminophenazine used in the treatment regimen for MDR-TB. Hematological association with clofazimine includes hemolytic anemia, anemia, reduction in total red cell number, macrocytosis, poikilocytosis, hypochromia, normocytic and normochromic anemia, anisocytosis and elevated reticulocyte counts [78,79].

### 5.11. Meropenem/Imipenem

The carbapenem group of drugs, which includes meropenem, imipenem and ertapenem, is currently used to treat MDR and extensively drug-resistant tuberculosis (XDR-TB), with imipenem–cilastatin and meropenem listed as add-on drugs in the updated WHO treatment guidelines [80].

There are limited available data regarding hematological adverse events due to carbapenems. The most frequently reported side effects of meropenem are thrombocytopenia, neutropenia and hemolytic anemia. Only one case of immune thrombocytopenia is available in the literature [81], and this case was due to interaction with PLTs or antibodies, thereby accelerating PLT clearance [82]. Neutropenia was described in a pediatric case; the mechanism primarily assumed to be involved was immune-mediated or direct toxicity in bone marrow, but other mechanisms were proposed, including hapten formation, complement-mediated cell destruction, antibody-mediated cell destruction and direct toxicity of myeloid precursors [83]. Data on hemolytic IgM drug-dependent anemia are available. Serum antibodies of a patient activated the complement system and induced intravascular lysis, with activation of the complement system [84].

Hematological side effects of imipenem are less frequently reported than are those for meropenem, accounting for neutropenia and thrombocytopenia [85,86].

### 5.12. Delamanid

Delamanid, a nitroimidazo-oxazole derivative, is a new anti-TB drug that exhibits potent in vitro and in vivo antitubercular activity against drug-susceptible and -resistant strains of *Mycobacterium tuberculosis*. It is approved in several countries, including Japan and various countries in Europe, for use as part of an appropriate combination regimen in adults with MDR-TB when an effective treatment regimen cannot otherwise be composed due to resistance or tolerability. Despite the presence of hematological side effects on the warning label, no data in the literature are available.

### 5.13. Thioamides

Thioamides, including ethionamide and prothionamide, are considered interchangeable second-line bacteriostatic agents for MDR-TB. Only two cases of agranulocytosis during therapy for pulmonary tuberculosis and a case of leuko-neutropenia were reported in the literature [87].

### 5.14. Linezolid

Linezolid was recently reclassified by the World Health Organization (WHO) as a group A drug for the treatment of MDR-TB and XDR-TB, suggesting that it should be included in the regimen for all patients unless contraindicated [88].

Linezolid’s use carries a considerable risk of toxicity, with the optimal dose and duration remaining unclear. Prolonged linezolid treatment (>14 days) is considered to increase the risk of hematological adverse events, including anemia, leucopenia, thrombocytopenia and even pancytopenia [88]. There are case reports of reversible thrombocytopenia, anemia and neutropenia associated with linezolid therapy. In phase III studies, 2.4% of patients treated with linezolid and 1.5% of patients treated with comparator drugs developed reversible thrombocytopenia, but there is no evidence of an increased risk of agranulocytosis, aplastic anemia or other irreversible blood dyscrasias [89]. The mechanism is unknown, but it is supposed that linezolid has a reversible suppressive effect on bone marrow [90,91]. Thrombocytopenia is among the most important adverse effects of linezolid treatment, with an incidence that varies considerably, although it has been associated with impaired renal function. The most common mechanism of thrombocytopenia associated with linezolid is the inhibition of the formation of platelets, while impaired renal function increases thrombocytopenia via a pharmacokinetic mechanism; therefore, linezolid doses should be reduced [92].

Kim et al. reported prolonged treatment, chronic liver disease and increased plasma linezolid concentration as risk factors associated with linezolid-induced thrombocytopenia [93].

Ring sideroblasts were found in the bone marrow of a patient who developed sideroblastic anemia after two months of therapy for abscesses due to MRSA [94]. Abundant ring sideroblasts in the bone marrow, thrombocytopenia and abnormal erythroblast morphology were found in a patient who developed renal dysfunction after 2 weeks of linezolid treatment [95]. Willekens et al. found that ring sideroblasts were in approximately 15% of patients who required bone marrow evaluation after moderate exposure to treatment of 13.5 days; the mechanism probably involved is linezolid’s specific binding to mitochondrial ribosomes, leading to inhibition of mitochondrial protein synthesis [96]. Linezolid may also cause myelosuppression and pancytopenia. Myelosuppression was observed as a side effect, which appeared to be both time and dose dependent and reversible upon cessation of the drug [97].

The risk of hematological adverse effects of linezolid may increase with prolonged use, as suggested by Letswee et al., who carried out a retrospective review of 27 patients hospitalized for MDR-TB and treated with the use of long-term linezolid (12 months); seven of these patients presented with hematological adverse effects that appeared after an average of 30 days of treatment [98]. The most common adverse effects were anemia and macrocytosis managed by dose reduction and blood transfusions. The authors recommended hematological monitoring weekly and thereafter monthly to detect early adverse effects, particularly for patients living with HIV [90,99]. In a systematic review on the efficacy and tolerability of linezolid, authors confirmed that linezolid is a viable option in the treatment of MDR/XDR TB, although patients ought to be monitored closely for the incidence of major adverse events, such as myelosuppression and neuropathy [100].

### 5.15. Bedaquiline

In 2012, the Food and Drug Administration approved the use of bedaquiline fumarate as part of combination therapy for MDR TB, and it dramatically improved the therapeutic success rate and presented great adherence to treatment. No hematological adverse events have been associated with bedaquiline.

## 6. Conclusions and Recommendations

In our literature review, we found how hematological events in TB patients can be predictors of both diagnosis and worse outcome for TB, regardless whether it is pulmonary, extra pulmonary or miliary. 

Even anti-TB therapies can cause hematological adverse events, among which some are serious and rare and compromise the patient’s recovery following treatment completion. In fact, if, on the one hand, there seems to be an underreporting of data with several case reports describing adverse events or rare presentations, in the absence of multicentric and prospective studies, it seems understandable; however, on the other hand, antituberculous therapy, faced with the challenge of confronting a big killer, can also effect the hematological profile. Therefore, a comprehensive approach to TB is needed.

Based on the literature review, we recommend the management of hematological manifestations and adverse drug events during treatment in active TB to be conducted as follows:-Hematological screening and follow-up, including CBC and coagulation, are always necessary both at the diagnosis of TB and during antitubercular treatment in order to monitor hematological parameters, even if rare, aiming at early detection of those factors predictive of therapeutic failure or worse outcome [101]; we suggest controlling them every fortnight in the first two months of antitubercular treatment and once a month in the following months.-Close monitoring of drug interactions and hematological adverse events is always recommended.-Short therapy regimens for MDR-TB, when possible, may also be useful to reduce hematological toxicity, especially when this cannot be monitored.-Where possible, a CT scan with intravenous contrast medium administration should be carried out to rule out pulmonary embolism in high-risk TB patients.-Hematological disorders in patients with TB are possible in both young and old females and males with TB, but they seem more frequent in the elderly. This observation underlines that the elderly are a particularly vulnerable group with a high risk of poor outcome who need careful medical attention and hematological monitoring, even during antituberculous treatment [102].-Drug discontinuation in cases of serious adverse events is always necessary. In the case of linezolid, consider starting again with a lower dose (300 mg/day instead of 600 mg/day) if the myelosuppression resolves and if linezolid is considered essential for the regimen [101]. Other non-drug-related causes must also be considered.-The development of a comprehensive prospective database on hematological manifestations in patients with TB, especially rare ones, is important; this must include hematological adverse reactions during antituberculous treatment to monitor and evaluate the severity and the impact on TB outcome and to improve the control of the treatment.

## Figures and Tables

**Table 1 microorganisms-09-01477-t001:** Frequency of hematologic patterns in TB patients.

Hematologic Findings	Pulmonary TB	Extrapulmonary TB	Miliary/Disseminated TB
Anemia	Common	Common	Common
Thrombocytopenia	Rare	Rare	Common
Thrombocytosis	Common	Common	Uncommon
WBC abnormalities	Common	Common	Common
Pancytopenia	Uncommon	Uncommon	Common
Henoch–Schönlein purpura	Rare	No data	No data
Leukemoid reaction	No data	Rare	No data
Hemolytic anemia	No data	No data	Rare
Hemophagocytic lymph histiocytosis	No data	No data	Rare
Disseminated intravascular coagulation	Rare	No data	Rare
Thromboembolism	Uncommon	No data	No data

TB: tuberculosis; WBC: white blood cells.

**Table 2 microorganisms-09-01477-t002:** Common and rare hematologic adverse event due to TB treatment.

Drugs	Adverse Events
Rifampicin	Thrombocytopenia, thrombocytopaenic purpura, leukopenia, disseminated intravascular coagulation, agranulocytosis, hemolysis
Isoniazid	Red cell aplasia, sideroblastic anemia, agranulocytosis, thrombocytopenia
Ethambutol	Hemolytic anemia, neutropenia and eosinophilia, agranulocytosis, thrombocytopenia
Pyrazinamide	Thrombocytopenia, megaloblastic anemia, sideroblastic anemia
Rifapentin	Acute porfiria, aplastic anemia, autoimmune hemolysis, agranulocytosis, leucemoid reaction, leukopenia, thrombocytopenia, TTP
Rifabutin	Leukopenia
Cycloserine	Vitamin B12 deficiency, folic acid deficiency, megaloblastic anemia, sideroblastic anemia
Para-Aminosalicylic Acid (PAS)	Agranulocytosis, thrombocytopenia, hemolytic anemia, methemoglobinemia
Streptomycin	Eosinophilia, leukopenia, thrombocytopenic purpura
Amikacin	DRESS syndrome
Levofloxacin	Thrombocytopenia, pancytopenia, hemolytic anemia
Moxifloxacin	Neutropenia
Clofazimine	Hemolytic anemia, anemia, reduction in total red cell number, macrocytosis, poikilocytosis, hypochromia, normocytic and normochromic anemia, anisocytosis, elevated reticulocyte counts
Ethionamide Prothionamide	Agranulocytosis, leuko-neutropenia
Delamanid	No data
Meropenem/Imipenem	Thrombocytopenia, neutropenia, hemolytic anemia
Linezolid	Thrombocytopenia, anemia and neutropenia, myelosuppression

TPP: thrombotic thrombocytopenic purpura; PAS: para-aminosalicylic acid; DRESS: drug rash with eosinophilia and systemic symptoms.

## Data Availability

Our data are available on PubMed, Scopus, Google Scholar, EMBASE, Cochrane Library and WHO websites (http://www.who.int accessed on 20 May 2021) in accordance with bibliography references.

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
