# Peer review of "Common and Rare Hematological Manifestations and Adverse Drug Events during Treatment of Active TB: A State of Art"

_microorganisms, 2021, doi:10.3390/microorganisms9071477_

Round 1
Reviewer 1 Report
The manuscript by Minardi et al. presents a literature review of known hematological manifestations and adverse drug events during treatment in active tuberculosis. The topic is of great interest for the specialists in the fields of tuberculosis and hematology; the review gives a short but comprehensive overview of known adverse events and provides clinical recommendations for monitoring and preventing some of them. Although this is an important review for the field, the manuscript in its current state lacks coherency in presentation of the data and in format. I would suggest one or two main authors to go through the whole text once again and bring the text to the same format to make it more readable and easier to follow. Also, a revision in terms of language use is highly recommended.
Please use a coherent format: not capitalizing names of diseases and drugs, using abbreviations throughout the text after you introduced it for the first time, not introducing abbreviations several times (see lines 401 or 461 as an example).
Also, here are some more detailed comments by sections. However, the revision of the manuscript should not be limited by addressing these comments.
Introduction
Please check the format of references
Lines 44, 189, 403: please use the internationally accepted format of writing the name of the bacterial species: Mycobacterium tuberculosis.
Materials and Methods
I believe you should state the exact moment when the search was done, for example from 1950 to May 2021.
Results
Please check the format of references: lines 65, 124, 338 and others
Lines 137-140, 244-247, 273-275: please check the sentence
Line 238: “miliary”
Generic names of drugs should not be capitalized (isoniazid, rifampicin etc)
In the section 5 please use a coherent format for each sub-section. For some drugs you give some short description of structure and mechanism of action while for others you don’t. You even provide some history for streptomycin but not for other drugs.
Tables 1 and 2: please use the coherent format for each cell
Conclusion
The effort made by the authors to give some practical recommendations to monitor and prevent possible hematological adverse events is much appreciated. However, I would recommend to revise this section as it is hard to follow sometimes – possibly, due to language use. Also, line 480: I suppose the authors recommend measures, not manifestations and adverse events.
Author Response
We have appreciated the positive feedback to our manuscript “Common and rare hematological manifestations and adverse drug events during treatment in active TB: a state of art”. We have considered all the useful suggestions made by the referees and we have implemented the text. We have also satisfied the technical requirements according to the journal guidelines. Modifications have been highlighted using the "track changes" feature. Also, a native English speaker has been engaged to improve the fluency and the readability of the manuscript.
We believe that the revision proposed by the referees, and further implemented in the text, contributed to improve the manuscript. Thus, we kindly ask you to re-consider the manuscript for publication.
Please find a point-by-point response to the referees’ comments below.
Best regards,
Dr. Francesco Di Gennaro
REVIEWER 1
The manuscript by Minardi et al. presents a literature review of known hematological manifestations and adverse drug events during treatment in active tuberculosis. The topic is of great interest for the specialists in the fields of tuberculosis and hematology; the review gives a short but comprehensive overview of known adverse events and provides clinical recommendations for monitoring and preventing some of them. Although this is an important review for the field, the manuscript in its current state lacks coherency in presentation of the data and in format. I would suggest one or two main authors to go through the whole text once again and bring the text to the same format to make it more readable and easier to follow. Also, a revision in terms of language use is highly recommended.
Response:
We thank you very much for the encouraging feedback on our manuscript. We followed your suggestions and now we believe that the paper is more usable for the scientific community.
Please use a coherent format: not capitalizing names of diseases and drugs, using abbreviations throughout the text after you introduced it for the first time, not introducing abbreviations several times (see lines 401 or 461 as an example). Also, here are some more detailed comments by sections. However, the revision of the manuscript should not be limited by addressing these comments.
Response: Thank you a lot for your suggestions. We modified following your comments.
Introduction
Please check the format of references.
Lines 44, 189, 403: please use the internationally accepted format of writing the name of the bacterial species: Mycobacterium tuberculosis.
Response: Thank you, we modified following your indications
Materials and Methods
I believe you should state the exact moment when the search was done, for example from 1950 to May 2021.
Response: We modified Material and methods as follow: “We conducted a research on PubMed, Scopus, Google Scholar, EMBASE, Cochrane Library and WHO websites (http://www.who.int) starting from 1950 to May 2021, looking for articles referring about interaction between TB and common or rare hematological manifestation. We included all articles dealing with epidemiology, physiopathology, risk factors, clinical features, screening and diagnosis, treatment and management”.
Results
Please check the format of references: lines 65, 124, 338 and others
Response: Thank you, we check the format of references
Lines 137-140, 244-247, 273-275: please check the sentence
Response: Thank you, we checked and modified the sentences as you rightly suggested.
Line 238: “miliary”
Response: Sorry for this mistake, we corrected the word miliary
Generic names of drugs should not be capitalized (isoniazid, rifampicin etc). In the section 5 please use a coherent format for each sub-section. For some drugs you give some short description of structure and mechanism of action while for others you don’t. You even provide some history for streptomycin but not for other drugs.
Response: Thank you for your suggestions. According to your revision, we added for each drugs a short description.
Tables 1 and 2: please use the coherent format for each cell
Response: Thank you, we used the same format for tables 1 and 2
Conclusion
The effort made by the authors to give some practical recommendations to monitor and prevent possible hematological adverse events is much appreciated. However, I would recommend to revise this section as it is hard to follow sometimes – possibly, due to language use. Also, line 480: I suppose the authors recommend measures, not manifestations and adverse events.
Response: Thank you a lot for your suggestions. We modified that section in “Based on literature review we recommend to manage hematological manifestations and adverse drug events during treatment in active TB as follow:
- hematological screening and follow-up, including CBC and coagulation, is always necessary both at the diagnosis of TB and during antitubercular treatment in order to monitor hematological parameters, even if rare, aiming at early detection of those factors predictive of therapeutic failure or worse outcome [88]; we suggest to control them every fortnight in the first two months of antitubercular treatment and once in a month in the following months;
- close monitoring of drug interactions and hematological adverse events is always recommended;
- short therapy regimens for MDR-TB, when possible, may also be useful to reduce hematological toxicity, especially when this cannot be monitored;
- where possible, a CT scan with intravenous contrast medium administration should be done to rule out pulmonary embolism in high risk TB patients;
- hematological disorders in patients with TB are possible in both young and old women and men with TB, but they seem more frequent in the elderly. This observation underlines that the elderly are a particularly vulnerable group with a higher risk of poor outcome and who need more medical attention and hematological monitoring even during anti-tuberculous treatment[89];
- Drug discontinuation in case of a serious adverse event is always necessary. In the case of linezolid, consider starting again with a lower dose (300 mg/day instead of 600) if the myelosuppression resolves and if linezolid is considered essential for the regimen [88]. Other non-drug related causes must also be considered;
- the development of a comprehensive prospective database on hematological manifestations in patients with TB, especially rare ones, it is important; this must include hematological adverse reactions during anti-tuberculous treatment to monitor and evaluate the severity and the impact on TB outcome and improve the control of the treatment. ”

Reviewer 2 Report
The manuscript deals with a searched PubMed, Scopus, Google Scholar, EMBASE, Cochrane Library and WHO websites from 1950 to date for papers on the interaction between Tuberculosis and common and rare hematological manifestations.
The survey has been well conducted with exhaustive collected data, and the results are sound and of clinical interest.
Main conclusion seems to be important, i.e. "hematological screening and follow-up including blood count haemochrome and coagulation is always necessary both at the diagnosis of Tuberculosis and during antitubercular treatment in order to monitor hematological parameters"
Only minor concerns must be addressed:
- Streptomicin must be Streptomycin both in text and table 2
- No Author Contributions have been indicated.
- Data Availability: A site where data supporting reported results can be found, including links to publicly archived datasets analyzed or generated during the study should be provided.
Author Response
We have appreciated the positive feedback to our manuscript “Common and rare hematological manifestations and adverse drug events during treatment in active TB: a state of art”. We have considered all the useful suggestions made by the referees and we have implemented the text. We have also satisfied the technical requirements according to the journal guidelines. Modifications have been highlighted using the "track changes" feature. Also, a native English speaker has been engaged to improve the fluency and the readability of the manuscript.
We believe that the revision proposed by the referees, and further implemented in the text, contributed to improve the manuscript. Thus, we kindly ask you to re-consider the manuscript for publication.
Please find a point-by-point response to the referees’ comments below.
Best regards,
Dr. Francesco Di Gennaro
REVIEWER 2
The manuscript deals with a searched PubMed, Scopus, Google Scholar, EMBASE, Cochrane Library and WHO websites from 1950 to date for papers on the interaction between Tuberculosis and common and rare hematological manifestations.
The survey has been well conducted with exhaustive collected data, and the results are sound and of clinical interest.
Main conclusion seems to be important, i.e. "hematological screening and follow-up including blood count haemochrome and coagulation is always necessary both at the diagnosis of Tuberculosis and during antitubercular treatment in order to monitor hematological parameters"
Response: We thank you very much for the encouraging feedback on our manuscript.
Only minor concerns must be addressed:
- Streptomicin must be Streptomycin both in text and table 2
- No Author Contributions have been indicated.
- Data Availability: A site where data supporting reported results can be found, including links to publicly archived datasets analyzed or generated during the study should be provided.
Response: Thank you a lot for your comments. We added author’s contributions and data availability. Furthermore, native English speaker revised the manuscript and corrected “Streptomicin” in “Streptomycin” both in the text and in the tables. We agree with your suggestions, thanks to which we believe the paper has improved considerably.

Round 2
Reviewer 1 Report
The authors addressed most of my comments. The manuscript is now more coherent and easier to follow.
There are still some formatting issues detected (introducing an abbreviation but not using it later, capitalising terms that shouldn't be capitalised etc.) but I leave it to the editor's discretion.